# REAC Antalgic Neuro Modulation in Chronic Post Herpetic Neuralgia

**DOI:** 10.3390/jpm13040653

**Published:** 2023-04-11

**Authors:** Ana Rita Pinheiro Barcessat, Lucas dos Santos Nunes, Rebeca Góes Gonçalves, Danyela Darienso

**Affiliations:** 1Health and Biological Sciences Department, Federal University of Amapá, Macapá 68900-350, Brazil; 2Department of Biomedical Sciences, University of Sassari, 07100 Sassari, Italy; 3Health Science Post Graduate Program-PPGCS–UNIFAP, Federal University of Amapá, Macapá 68900-350, Brazil

**Keywords:** pain, postherpetic neuralgia, neurostimulation, neuromodulation, neurobiological modulation, varicella-zoster virus

## Abstract

Background: Chronic post-herpetic neuralgia (CPHN) is a symptomatic condition that afflicts adults and elderly individuals. The chronicity of this symptomatology can be conditioned by the epigenetic modifications induced by the virus on the processes of neurotransmission and sensitivity to pain. The aim of this study is to investigate whether manipulating endogenous bioelectrical activity (EBA), responsible for neurotransmission processes and contributing to the induction of epigenetic modifications, can alleviate pain symptoms. Methods: This manipulation was carried out with the antalgic neuromodulation (ANM) treatment of radioelectric asymmetric conveyer (REAC) technology. Pain assessment before and after treatment was performed using a numerical analog scale (NAS) and a simple descriptive scale (SDS). Results: The results of the analysis showed an over four-point decrease in NAS scale score and over one point decrease in SDS scale score, with a statistical significance for both tests of *p* < 0.005. Conclusions: The results obtained in this study demonstrate how REAC ANM manipulation of EBA can lead to improvement in epigenetically conditioned symptoms such as CPHN. These results should prompt further research to expand knowledge and ensure optimized therapeutic outcomes.

## 1. Introduction

Varicella-zoster virus (VZV) is a type of human neurotropic virus that belongs to the alpha herpesvirus family [1]. The first exposure to the VZV virus typically results in the development of chickenpox, also known as varicella. Chickenpox is a highly contagious disease that is characterized by the appearance of itchy blisters on the skin, along with fever and general malaise. Once the clinical course of chickenpox has ended, the virus remains dormant in the nervous system for the life of the infected individual, specifically in the cranial nerve, dorsal root, and autonomic ganglia [2].

VZV can be reactivated even after many years. The mechanisms that determine both the latency and reactivation of VZV are not yet fully understood, but research has shown that they may be linked to epigenetic mechanisms [3]. These modifications can influence the activity of VZV genes and affect the ability of the virus to reactivate from its latent state.

Reactivation from latent VZV results in a clinical and symptomatic picture known as herpes zoster or shingles [4,5,6].

Shingles is a painful rash consisting of blisters that typically form within 7–10 days and resolve completely within 2–4 weeks. However, in 5–30% of patients [7], the pain associated with the lesions induced by HZ does not end with the disappearance of the skin rash and persists chronically [8], with little or no responsiveness to pain-relieving drugs [9]. This clinical situation is defined as chronic postherpetic neuralgia (CPHN) [10] and represents one of the most common forms of neuropathic pain.

Neuropathic pain is a multifaceted condition that results from maladaptive structural changes and a wide range of cell–cell interactions and molecular signaling pathways. This leads to the sensitization of nociceptive pathways, which can cause debilitating pain. These changes can involve various mechanisms, including alterations in ion channels, glial-derived mediators, and epigenetic regulation, among others [11].

In this study, we investigated the efficacy of a specific neurobiological solitonic modulation treatment called antalgic neuromodulation (ANM) [12]. ANM is a specific treatment approach that utilizes radioelectric asymmetric conveyor (REAC) technology to combat neuropathic pain. This innovative strategy has been developed to target the underlying mechanisms of neuropathic pain.

The aim of this study was to assess whether ANM could effectively reduce pain in patients with CPHN, which can significantly impact their quality of life.

The findings of this study suggest that the use of REAC technology can be useful in the treatment of CPHN.

## 2. Materials and Methods

### 2.1. Power Analysis

The statistical sample size was calculated using GPower software version 3.1.9.4, with the following parameters: effect size of 0.60, alpha probability error of 0.05, power of 0.90, and using the Wilcoxon signed-rank test. Based on these parameters, the sample size calculation yielded a sample size of 27 subjects, which was deemed sufficient to detect statistically significant differences between groups.

### 2.2. Inclusion and Exclusion Criteria

The inclusion criteria for this study were as follows: men and women aged over 45 years, with a previous diagnosis of herpes zoster and CPHN symptoms present for more than 120 days, with poor or absent sensitivity to antiviral, anti-inflammatory, pain-relieving treatments, and anticonvulsant drugs.

The exclusion criteria were concomitant neurological and rheumatological diseases, diabetes, fibromyalgia, psychiatric diseases, and tumor pathologies.

### 2.3. Population

This study is part of a project aimed at improving the quality of life of the population of the Brazilian region of Amapá. To ensure a humanitarian approach towards improving the quality of life of the local population, we chose to enroll a larger number of patients in our study than was calculated as necessary to establish the validity of our results. This decision reflects our commitment to maximizing the potential benefits of our research for the community, and to ensuring that as many individuals as possible have access to the latest advances in medical science.

In total, 53 subjects aged between 49 and 84 years, with an average of 65.30, of which 31 were male (mean age 64.25) and 22 were female (mean age 66.70) were enrolled between July 2021 and October 2022. The study population was multiracial, and all participants completed the course of therapy.

### 2.4. Pain Assessments before Treatment

All patients who participated in the study perceived and reported pain essentially localized in the thoracic area along the ribs.

Before receiving the treatment (T0), all patients were asked to rate their pain by completing a numerical analog scale (NAS) with a value from 0 to 10, where 0 represents no pain and 10 represents extreme pain [13]. Patients were also asked to classify their pain using a simple descriptive scale (SDS) [14,15] with values from 0 to 4, where 0 represents the absence of pain, 1 mild pain, 2 moderate pain, 3 severe pain, and 4 extreme pain.

To compare the two scales, we aligned the initial and final values, with 0 as no pain and 10 as severe-extreme pain, while for the intermediate values, we classified the NAS from 1 to 3 as medium pain, from 4 to 6 as moderate pain, and from 7 to 9 as severe pain.

Immediately after collecting the NAS and SDS data, the patients underwent the first ANM treatment (T1).

The follow-up (T2) was set at about three months after the end of the treatment cycle. This interval was considered necessary to allow biological time necessary for the remodulation of the epigenetic modifications [16] that affected the chronicity/intensity of CPHN.

### 2.5. REAC Technology

Radio Electric Asymmetric Conveyer (REAC) technology is a relatively new medical noninvasive technology. It was developed by two Italian researchers Salvatore Rinaldi and Vania Fontani. The REAC technology uses radio electric fields that are asymmetrically conveyed to the body through an asymmetric conveyer probe (ACP).

These radio electric fields, using specific administration protocols, are designed to promote the progressive recovery and optimization of the natural endogenous bioelectric activity (EBA) [17,18] that is produced by the cells and tissues, to promote various healing and regeneration processes [19].

The effect of progressive improvement of the EBA is attributed to the solitonic effect [20,21], which is typically produced by the REAC technology.

A soliton is a type of wave that maintains its shape and amplitude as it propagates through a medium due to its self-reinforcing nature [22]. In the context of neurotransmission, the soliton effect describes the propagation of EBA signals along neurons in a manner similar to that of a soliton [23].

The soliton-like behavior of EBA signals is influenced by the electrical properties of the neuron membrane and the activity of voltage-gated ion channels [24]. When these properties are finely balanced by the REAC treatments, the EBA signal can exhibit soliton-like behavior, maintaining its shape and amplitude as it travels along the neuron. This behavior is thought to contribute to the rapid long lasting and efficient conduction of nerve impulses along long axons in the peripheral nervous system.

This effect has been demonstrated to persist even after a single administration of REAC treatment [25,26,27], indicating the potential for long-lasting therapeutic benefits in pain management [12,28,29].

### 2.6. Treatments

Antalgic neuro modulation (ANM) treatment is a solitonic neuromodulation treatment with REAC technology designed to remodel spontaneous spiking activity and the extracellular summation of postsynaptic potentials generated by neuronal population communication, i.e., the so-called local potential fields (LPFs) at the thalamic and cortical levels, particularly in chronic pain [12]. This effect is made possible by REAC technology designed to reorganize endogenous bioelectrical activity (EBA), which is also the basis of neurotransmission processes.

The administration of the ANM involves the positioning of a probe, called an asymmetric conveyer probe (ACP), on the patient’s back to cover the cervical, thoracic and lumbar tracts (Figure 1).

The ACP is held in place by an elastic tubular mesh and then connected to the device via a cable. Each treatment session lasts 15 min; the protocol specifies 1–3 treatments sessions per day during the week (Monday–Friday) depending on the logistical availability of the patients. A complete therapeutic cycle consisted of 18 sessions. The administration parameters are established by the manufacturer and cannot be modified by the operator. The device used is the BENE 110 (ASMED, Scandicci, Italy).

### 2.7. Ethics

This study was approved by the Ethical Comity of the Federal University of Amapá, Macapá, with approval code NO. 4.763.000. Since lack of responsiveness to previous treatments for CPHN was a component of the inclusion criteria, we believed, for ethical purposes, that patients could be free to continue taking the previous therapies at their discretion.

## 3. Results

### 3.1. NAS Results

All participants showed a reduction in NAS scores at T2 compared to T0 (Figure 2).

Of the 16 patients who at T0 with NAS had expressed pain at 10, 4 at T2 expressed 7; 6 at T2 expressed 6; 4 at T2 expressed 5; and 2 at T2 expressed 4. Of the 22 patients who at T0 with NAS had expressed pain at 9, 8 at T2 expressed 5, 9 at T2 expressed 4, and 3 at T2 expressed 3. Of the 13 patients who expressed pain at T0 with NAS, 6 at T2 expressed 6; 1 at T2 expressed 5; and 6 at T2 expressed 4.

The mean decreases in NAS scores between T0 and T2 was 4.41 points.

### 3.2. SDS Results

The analysis of the results obtained using the SDS, calculating the difference of the subjective pain value at T0 and at the T2 follow-up shows an overall improvement in all patients (Figure 3).

Of the 16 patients who at T0 with SDS had expressed pain at 4 (extremely severe), 4 at T2 expressed 3 (severe pain) and 12 at T2 expressed 2 (medium pain). Of the 37 patients who at T0 with SDS had expressed pain at 3 (severe pain), 26 at T2 expressed 2 (medium pain); 11 at T2 expressed 1 (medium pain).

The mean decrease in SDS score between T0 and T3 was 1.43 points.

No patients reported side effects or unwanted effects.

### 3.3. Results of Statistical Analysis

The collected data were incorporated into an Excel sheet and subsequently subjected to a statistical analysis using the Statistical Package for Social Science (SPSS) version 22.

The Wilcoxon and sign statistical tests were applied to evaluate differences in pain scores between the two time points for each of the chosen scales. For all the applied tests, a statistical significance *p* < 0.005 was found (Table 1).

## 4. Discussion

Chronic pain is an extremely disabling disorder that afflicts approximately two-thirds of the Brazilian population [30] and one-fifth of the European and US populations [31]. The majority of neuropathic pain conditions exhibit a paradoxical manifestation of both sensory loss and pain, which may be accompanied by sensory hypersensitivity in the affected region [11]. An important distinguishing feature in most neuropathic types of pain is the paradoxical combination of sensory loss and pain either with or without sensory hypersensitivity phenomena in the painful area [11].

The mechanisms relating chronic pain to altered neurotransmissions [32] and epigenetic modifications have been a topic of emerging interest for some years now [33]. This correlation arises from the fact that numerous neuronal processes, such as neurotransmission, neuroplasticity, and other neuronal functions, including the learning and memorization processes, may be conditioned by epigenetic processes [34,35,36].

Among the causes of chronic pain, neuropathic pain accounts for approximately 15–25% [37]. One of the most common neuropathic pains in late adults and the elderly is CPHN [38]. CPHN has evidence of having an important epigenetic component that affects both its initiation and its evolution [37,38]. Therefore, the current knowledge of the epigenetic component in CPHN has led to the search for treatments that can positively regulate this component to modify it and improve the clinical picture.

To functionally organize altered neurotransmissions and epigenetic dysregulations, the use of technologies is emerging, using various modes of production of electric [39] or electromagnetic fields [40,41], including REAC technology. The REAC technology was designed to regulate altered neurotransmissions and epigenetic dysregulations that can determine both functional and structural alterations, through the progressive improvement in EBA.

Functional alterations can condition the processes of neural electrometabolic activity and neurotransmission, resulting in dysfunctional circuits that can also generate reverberation of nociceptive perception. Specific treatments of REAC technology have been shown to be able to reorganize both neural electrometabolic activity and neurotransmission in a functionally positive way [25,26], even in painful conditions [12].

The findings of this study are highly encouraging, as they demonstrate that REAC ANM treatment led to a significant reduction in pain levels. Specifically, patients experienced a decrease of over four points on the Numeric Analog Scale (NAS) and a reduction of more than one point on the Self-Rating Depression Scale (SDS). Notably, these outcomes were achieved in individuals who had been experiencing symptoms for over 120 days, with limited or no response to traditional antiviral medications, anti-inflammatory therapies, pain-relieving treatments, and anticonvulsant drugs.

These results highlight the potential of REAC ANM as a viable alternative for the management of chronic post-herpetic neuralgia, particularly in patients who have failed to respond to conventional treatment approaches. By offering significant pain relief to patients who have been suffering for prolonged periods, this innovative technology could help improve their quality of life and overall well-being.

## 5. Conclusions

The results of this study provide compelling evidence that REAC ANM manipulation of EBA can effectively improve epigenetically conditioned symptoms, such as CPHN. These findings have significant implications for advancing our understanding of the potential benefits of this novel approach in the management of neuropathic pain.

In conclusion, the present study sheds new light on the potential of REAC ANM as a promising therapeutic approach for the treatment of chronic neuropathic pain. By encouraging further research, these results pave the way for the development of more effective treatments that can improve the lives of those living with this challenging condition.

## Figures and Tables

**Figure 1 jpm-13-00653-f001:**
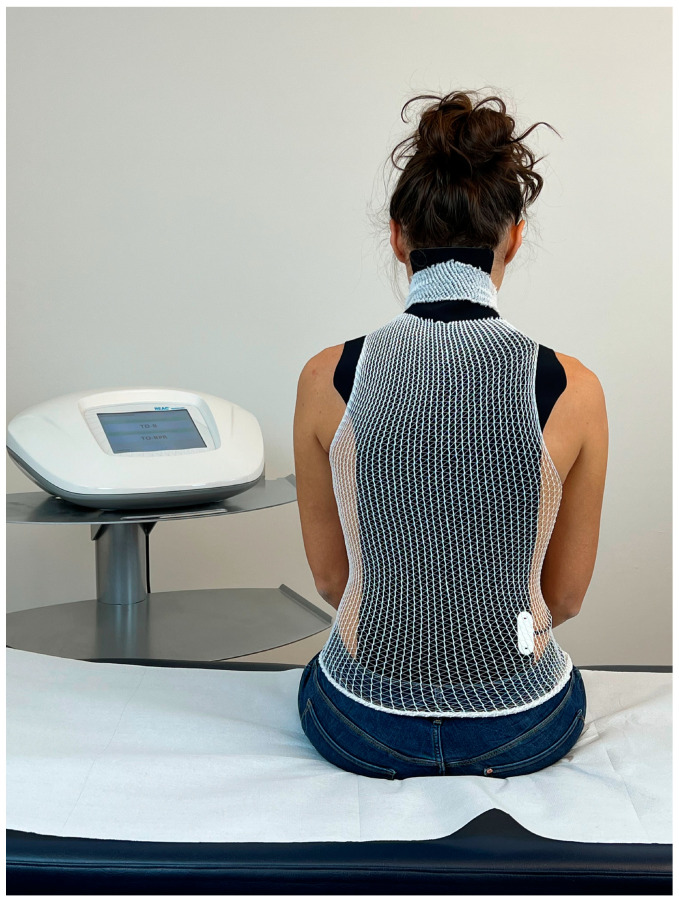
Example of the administration of REAC antalgic neuromodulation (ANM). The visible black part consists of the outer part of the asymmetric conveyer probe (ACP). White tubular elastic mesh holds the ACP in place. The ACP is connected with the device through a cable.

**Figure 2 jpm-13-00653-f002:**
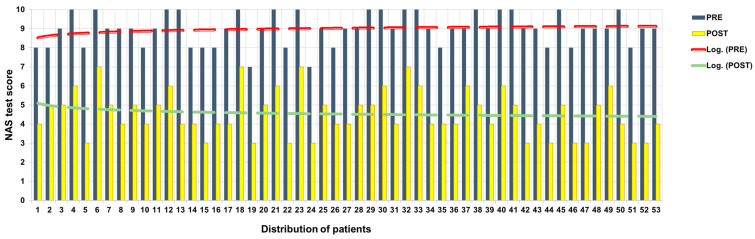
NAS score before and after ANM treatment.

**Figure 3 jpm-13-00653-f003:**
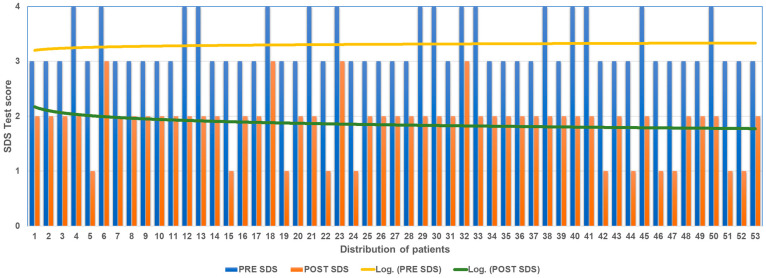
SDS score before and after ANM treatment.

**Table 1 jpm-13-00653-t001:** Wilcoxon and the Signs test for adequacy of the data.

	NAS Scale	SDS Scale
Wilcoxon test	Z = −6.429 Asymp. Sig. (2-tailed) = 0.000	Z = −6.547 Asymp. Sig. (2-tailed) = 0.000
Sign test	Z = −7.143 Asymp. Sig. (2-tailed) = 0.000	Z = −7.143 Asymp. Sig. (2-tailed) = 0.000

## Data Availability

All data are available at the Open Science Framework (https://osf.io/w5dz2/) URL accessed on 3 April 2023.

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
