# Peer review of "REAC Antalgic Neuro Modulation in Chronic Post Herpetic Neuralgia"

_jpm, 2023, doi:10.3390/jpm13040653_

Round 1

Reviewer 1 Report

The article entitled « REAC antalgic neuro modulation in chronic post herpetic neuralgia” is interesting. Nevertheless, I have some minor remarks to make.

Title: I propose to add “an open study” or something like that (not randomized with sham).

Line 57: It is mentioned of 52 patients which didn’t fit with 31 male and 22 females.

Line 65: I never heard of the SDS. Please add a reference or use a validated scale in Brazilian language.

Line 72: A delay of 3 months seems necessary to epigenetic modifications. This is an important data. Please add a reference.

Discussion:  authors need to open for a randomized study.

Author Response

Point to point answers.

Dear Editors and Reviewers,

first of all, thank you for your effort in reading and analyzing our manuscript to make it better than the initial draft.

Responses to the observations of the 1st Reviewer 2

  • Title: I propose to add “an open study” or something like that (not randomized with sham).

Answer: Thanks for the proposal.

  • Line 57: It is mentioned of 52 patients which didn’t fit with 31 male and 22 females.

Answer: Thank you for noticing this discrepancy, which occurred due to a typo. We corrected fifty-two with fifty-three.

  • Line 65: I never heard of the SDS. Please add a reference or use a validated scale in Brazilian language.

Answer: The simple descriptive pain scale (SDS) was formulated by Keele in 1948 using 4-5 descriptive levels (Reville SI, Robinson JO, Rosen M, Hogg MI: The Reliability of linear Analogue Scale for Evaluation of Pain. Anaesthesia: 36:186-187, 1977)

As requested, we have included two references:

  • Sriwatanakul, K.; Kelvie, W.; Lasagna, L.; Calimlim, J.F.; Weis, O.F.; Mehta, G. Studies with different types of visual analog scales for measurement of pain. Clin Pharmacol Ther 1983, 34, 234-239, doi:10.1038/clpt.1983.159.

  • George, S.; Ang, H.-G. Comparison of two pain rating scales among Chinese cancer patients. Chinese Medical Journal 1992, 105, 953-956, doi: doi:10.5555/cmj.0366-6999.105.11.p953.01.

  • Line 72: A delay of 3 months seems necessary to epigenetic modifications. This is an important data. Please add a reference.

Answer: At the moment and in our knowledge, there is no specific period of time in which we can be sure that we have brought about an epigenetic modification.

However, we know that epigenetic changes can also occur in a short time such as for example during the menstrual cycle (Munro SK, Farquhar CM, Mitchell MD, Ponnampalam AP. Epigenetic regulation of endometrium during the menstrual cycle. Mol Hum Reprod. 2010 May;16( 5):297-310.doi:10.1093/molehr/gaq010.Epub 2010 Feb 5. PMID:20139117.)

The fact of waiting at least three months from the end of the treatments before carrying out the follow up arises from an empirical experience. We have however added the following reference: Harrison, A.; Parle-McDermott, A. DNA methylation: A timeline of methods and applications. Front Genet 2011, 2, 74.

  • Discussion: authors need to open for a randomized study

Answer: we take the liberty of pointing out that the phrase we used "These results should prompt further research to expand knowledge and ensure optimized therapeutic outcomes" opens up to any type of study, even randomized double-blind studies. So, we think it doesn't preclude any kind of other future studies.

Reviewer 2 Report

Barcessat et al report the use of REAC  dioelectric asymmetric conveyor (REAC) technology called antalgic 4neuromodulation (ANM) in reducing the pain of patients suffering from

 Chronic post-herpetic neuralgia (CPHN). The results seems interesting and convincing that indeed REAC technology can provide relief to these patients.

·         The authors positioned the probe, ACP, on the patient's back to cover the cervical, thoracic, and lumbar tracts. Was the pain experienced by the patients global or localized in specific somatic areas?

·         The authors mention changes in neuronal activity as the basis of epigenetic processes throughout the manuscript. Neuronal bioelectrical activity may or may not induce the epigenetic process and epigenetic changes may or may not be induced due to changes in neuronal electrical activity. Hence the statement is factually incorrect.

·         Does these patients report pain? Is it localized or global?

·         Please provide more information on REAC technology in the introduction. And provide a review of the published literature that has reported the use of this technology for other neurological disorders

·         For Figure 2 and Figure 3: what does the log conversion reflect?

·         Was there any reported side effects or negative effects of this treatment? 

Author Response

Dear Editors and Reviewers,

First of all, thank you for your effort in reading and analyzing our manuscript to make it better than the initial draft.

Responses to the observations of the  Reviewer 2

  • The authors positioned the probe, ACP, on the patient's back to cover the cervical, thoracic, and lumbar tracts. Was the pain experienced by the patients global or localized in specific somatic areas?

Answer: In Pain assessments before treatment, we have added the following sentence:

All patients who participated in the study perceived and reported pain essentially localized in the thoracic area along the ribs.

  • The authors mention changes in neuronal activity as the basis of epigenetic processes throughout the manuscript. Neuronal bioelectrical activity may or may not induce the epigenetic process and epigenetic changes may or may not be induced due to changes in neuronal electrical activity. Hence the statement is factually incorrect.

Answer: Dear reviewer, thank you for your observation, with which we fully agree, but we are unable to understand from which sentences of the manuscript it originated.

We think we have expressed that endogenous bioelectrical activity (EBA) is the basis of epigenetic processes, both in the abstract and in the discussion of the manuscript, also citing the appropriate bibliographic entries.

In the abstract we have write: The purpose of this study is to examine whether manipulation of endogenous bioelectrical activity (EBA), which is the basis of epigenetic processes and neurotransmission, can lead to an improvement in pain symptoms.

In discussion we have write: This correlation arises from the fact that numerous neuronal processes, such as neurotransmission, neuroplasticity, and other neuronal functions, including learning and memorization processes, are conditioned by epigenetic processes.

  • Does these patients report pain? Is it localized or global?

Answer: In Pain assessments before treatment, we have added the following sentence:

All patients who participated in the study perceived and reported pain essentially localized in the thoracic area along the ribs.

  • Please provide more information on REAC technology in the introduction. And provide a review of the published literature that has reported the use of this technology for other neurological disorders.

Answer: Thank you for your suggestion.

As suggested, we have provided more information on REAC technology and also a brief overview of its fields of application with related bibliography.

However, we thought it more appropriate to include this integration in materials and methods:

REAC Technology

Radio Electric Asymmetric Conveyer (REAC) technology is a relatively new medical noninvasive technology. It was developed by two Italian researchers Salvatore Rinaldi and Vania Fontani. The REAC technology uses radio electric fields that are asymmetrically conveyed to the body through an asymmetric conveyer probe (ACP).

These radio electric fields, using specific administration protocols, are designed to promote the recovery and optimization of the natural endogenous bioelectric activity that is produced by the cells and tissues, to promote various healing and regeneration processes.

REAC technology, using specific administration protocols, has been shown to have a range of therapeutic applications. Clinical studies have demonstrated the effectiveness of REAC treatment protocols in mood and behavior disorders even in developmental age, motor control disorders, pain, and reparative and regenerative medicine.

  • For Figure 2 and Figure 3: what does the log conversion reflect?

Answer: A logarithmic trendline which is useful when the rate of change in the data increases or decreases rapidly.

  • Was there any reported side effects or negative effects of this treatment?

Answer: At the end of the results section, we have added the following sentence:

No patients reported side effects or unwanted effects.

Round 2

Reviewer 2 Report

I am okay with authors' response except for this one. My suggestion is that authors should rephrase these sentences (written below) to suggest that EBA might induce epigenetic processes with relevant reference(s) and remove any such sentence that implies that EBA will (always) be the basis of epigenetic changes.

"In the abstract we have write: The purpose of this study is to examine whether manipulation of endogenous bioelectrical activity (EBA), which is the basis of epigenetic processes and neurotransmission, can lead to an improvement in pain symptoms.

In discussion we have write: This correlation arises from the fact that numerous neuronal processes, such as neurotransmission, neuroplasticity, and other neuronal functions, including learning and memorization processes, are conditioned by epigenetic processes."

Author Response

Dear Editor and Reviewers,

First of all, I would like to thank you again for your efforts to improve our manuscript.

Here are the point-to-point answers

Reviewer: My suggestion is that authors should rephrase these sentences (written below) to suggest that EBA might induce epigenetic processes with relevant reference(s) and remove any such sentence that implies that EBA will (always) be the basis of epigenetic changes.

"In the abstract we have write: The purpose of this study is to examine whether manipulation of endogenous bioelectrical activity (EBA), which is the basis of epigenetic processes and neurotransmission, can lead to an improvement in pain symptoms.

  • We have rephrased as follow:

The aim of this study is to investigate whether manipulating endogenous bioelectrical activity (EBA), responsible for neurotransmission processes and contributing to the induction of epigenetic modifications, can alleviate pain symptoms.

“In discussion we have write: This correlation arises from the fact that numerous neuronal processes, such as neurotransmission, neuroplasticity, and other neuronal functions, including learning and memorization processes, is conditioned by epigenetic processes."

  • We have rephrased as follow:

In discussion we have write: This correlation arises from the fact that numerous neuronal processes, such as neurotransmission, neuroplasticity, and other neuronal functions, including learning and memorization processes, may be conditioned by epigenetic processes."

Based on the content presented above, we are confident that we have effectively addressed all of the reviewer's comments and feedback.

Best regards
